# High diversity of coralline algae in New Zealand revealed: Knowledge gaps and implications for future research

Brenton A. Twist [1,2,3] *, Kate F. Neill[2], Jaret Bilewitch [2], So Young Jeong[4,5], Judy E. Sutherland[2], Wendy A. Nelson[2,6]

1 Institute of Marine Sciences, University of Auckland, Auckland, New Zealand, 2 National Institute of Water & Atmospheric Research, Wellington, New Zealand, 3 Department of Botany, University of British Columbia, Vancouver, British Columbia, Canada, 4 Department of Life Science, Chosun University, Dong-gu, Gwangju, Korea, 5 Griffith School of Environment and Australian Rivers Institute–Coast & Estuaries, Nathan Campus, Griffith University, Brisbane, Nathan, Queensland, Australia, 6 School of Biological Sciences, University of Auckland, Auckland, New Zealand

* brenton.twist@gmail.com

**Data Availability Statement:** All sequences are publicly available in GenBank (see S3 and S4 Tables for GenBank accession numbers). All other

## Abstract

Coralline algae (Corallinophycideae) are calcifying red algae that are foundation species in euphotic marine habitats globally. In recent years, corallines have received increasing attention due to their vulnerability to global climate change, in particular ocean acidification and warming, and because of the range of ecological functions that coralline algae provide, including provisioning habitat, influencing settlement of invertebrate and other algal species, and stabilising reef structures. Many of the ecological roles corallines perform, as well as their responses to stressors, have been demonstrated to be species-specific. In order to understand the roles and responses of coralline algae, it is essential to be able to reliably distinguish individual species, which are frequently morphologically cryptic. The aim of this study was to document the diversity and distribution of coralline algae in the New Zealand region using DNA based phylogenetic methods, and examine this diversity in a broader global context, discussing the implications and direction for future coralline algal research. Using three independent species delimitation methods, a total of 122 species of coralline algae were identified across the New Zealand region with high diversity found both regionally and also when sampling at small local spatial scales. While high diversity identified using molecular methods mirrors recent global discoveries, what distinguishes the results reported here is the large number of taxa (115) that do not resolve with type material from any genus and/or species. The ability to consistently and accurately distinguish species, and the application of authoritative names, are essential to ensure reproducible science in all areas of research into ecologically important yet vulnerable coralline algae taxa.

relevant data are within the manuscript and its Supporting Information files.

**Funding:** This research was funded by Ministry for Primary Industries (MPI) New Zealand under the Biodiversity Research Programme (ZBD2014-07), with further funding from National Institute of Water and Atmospheric Research (NIWA) SSIF funding. The funders had no role in study design, data collection and analysis, decision to publish, or preparation of the manuscript.

**Competing interests:** The authors have declared that no competing interests exist.

## Introduction

Coralline red algae (belonging to the orders Corallinales P.C.Silva & H.W.Johans., Hapalidiales W.A.Nelson, J.E.Sutherl., T.J.Farr & H.S.Yoon and Sporolithales L.Le Gall & G.W.Saunders) possess extra-cellular calcium carbonate, and are recognised as foundation species that are critical to the ecosystems in which they are found [1,2]. As calcified organisms, coralline algae are also vulnerable to the impacts of global climate change, such as warming seas and ocean acidification, and as such, have been receiving increased research attention over the last decade (e.g. [3–5]). Distributed from the poles to the tropics and occupying habitats from the intertidal zone through to the limits of the euphotic zone, these algae perform crucial ecological functions. Coralline algae provide structurally complex and food-rich habitats for small mobile invertebrates (e.g. [6]), support diverse and abundant faunal communities (e.g. [7]), influence the settlement of other macroalgae with both positive and negative interactions reported (e.g. [8,9]), induce settlement of a wide range of invertebrate species (e.g. [10–12]), and serve as seedbanks for microscopic algal life history stages (e.g. [13]). They also are major builders and stabilisers of reef framework in temperate and tropical regions (e.g. [1,4,14,15]) and play an important role in the recovery of biodiversity following disturbance (e.g. [8]).

Accompanying the surge of studies on coralline algal physiology, calcification, and responses to global and local anthropogenic stressors, over the past decade there has also been an increase in systematic research on corallines, which has resulted in an unprecedented number of discoveries, including the recognition of a new subclass, encompassing all coralline algal taxa (Corallinophycidae L.Le Gall & G.W.Saunders [16]), as well as new orders [16–18]. In addition to the new understanding of higher-level relationships, recent research on coralline algal systematics has clearly established that sequence data and phylogenetic analyses are essential for the characterisation of genera and species. This need for molecular sequence data has been articulated compellingly by multiple authors (e.g. [17,19–32]).

Reliable taxonomy, nomenclature and a robust phylogenetic framework are critical for all areas of research on coralline algae, with direct implications for the interpretation of physiological and ecological research, understanding the impacts of anthropogenic change and the provision of ecosystem services. This is particularly important as the induction of invertebrate settlement (e.g. [10,11]), growth rates (e.g. [11,33]), competitive ability (e.g. [34,35]), and responses to stressors (e.g. [36–38]) can all be species-specific. Assessing biodiversity at local and global scales, and understanding biogeography and distributional ranges requires reproducible and authoritative use of names and species concepts [19]. Based on investigations using DNA sequence data, strong caution has been expressed about basing the recognition of coralline algae (particularly non-geniculate/crustose species) on solely morphological and anatomical characters, and reports of morphologically identified non-geniculate coralline algae with geographically widespread and/or disjunct distributions have been called into question (e.g. [21,27,29,39]). While some species have been confirmed to have wide distributions (e.g. *Lithophyllum kaiseri* (Heydr.) Heydr., with widespread pan-tropical distribution [20], *Hydrolithon boergesenii* (Foslie) Foslie [40], and *Sporolithon indopacificum* Maneveldt & P.W.Gabrielson [21]), these appear to be the exception. For example, Adey et al. [31] found that in the cold-water Boreal region, the majority of species of *Phymatolithon* Foslie "are largely ecologically and biogeographically partitioned".

The aims of this study were to understand the diversity of coralline algae across the New Zealand region based on molecular data, and to estimate likely diversity that remains undiscovered based on current data and sampling approaches. Although this is a regional study, it has value beyond the SW Pacific. The resolution of generic and specific boundaries in the Corallinophycidae and the development of a robust phylogenetic framework for coralline algae

requires taxon sampling that reflects geographic and habitat diversity. These data provide new insights about the estimation of diversity and have implications for future research on coralline algae, their diversity, distribution and ecology.

## Materials and methods

### Specimen collection

A total of 796 samples of coralline algae were collected from 110 collection sites around southern New Zealand, extending earlier surveys of coralline algae in northern and central New Zealand (Fig 1 [41,42]). All collections were made under Special Permit 665 issued by the New Zealand Ministry for Primary Industries (MPI) which allows the taking of seaweed for the purposes of education and investigative research. Sampling was conducted in a range of different habitat types (e.g. rocky reefs, biogenic reefs, and soft bottoms) of varying exposures. Depending on accessibility, depth and available time, collections were made either by hand in the intertidal or subtidal, using a hammer and chisel, or by use of a dredge on soft bottom habitats in depths greater than 20 metres. At each site, a range of growth forms from different microhabitats were targeted for collection, in an attempt to representatively sample the biodiversity present. Specimens were examined using a dissecting microscope and sorted based on external morphological features (e.g. colour, reproductive features and growth form), depth and microhabitat. Wherever necessary, specimens were cleaned using forceps and a razor blade to remove fragments of rocky substrate underneath as well as epiphytes and invertebrates, prior to storage in silica gel for subsequent DNA analysis. Where possible, specimens were subsampled into formalin and subsequently rinsed in freshwater, prior to being transferred to an ethanol glycerol mix (1 glycerol: 7 ethanol 90%: 2 water) for preservation of anatomical features for long-term storage. Voucher specimens are currently housed at National Institute of Water and Atmospheric Research (NIWA) Wellington, and will be deposited in the Herbarium of the Museum of New Zealand Te Papa Tongarewa in Wellington (WELT [43]).

In addition to the sampling across the southern region, intensive sampling was undertaken at two sites to quantify diversity at small spatial scales, using two different approaches. At Butterfly Bay in Karitāne (Fig 1) sampling was undertaken over an area of approximately 0.02km$^2$, targeting a range of microhabitats (e.g. under macroalgal canopy, in crevices, on the surface of reef) from the intertidal zone to 10 metres below mean low water. In Moeraki (Fig 1), quantitative sampling was undertaken across a series of six boulders with little to no canopy cover at 1.5 metres below mean low water in a semi-sheltered sandy bay. Quantitative sampling was achieved using a line intercept transect method [44], by laying a transect across each boulder and recording the projected length of individual coralline alga (identified by taking a small sample for DNA analysis) beneath the transect to the nearest millimetre.

### DNA sequencing

DNA was extracted by grinding dried samples and then using the Qiagen DNeasy Blood and Tissue DNA Extraction Kit (Qiagen GmbH, Hilden) as per the manufacturers' instructions, except that the initial incubation of ground tissue was carried out with Buffer AL (200 µl), Buffer ATL (180 µl) and Proteinase K (20 µl) at 56°C for 2–4 hours.

We attempted to amplify and sequence the *psb*A marker from each DNA extract. Amplification and sequencing of the *rbc*L marker was subsequently carried out for a subset of these specimens based on the results of species delimitation analysis of the *psb*A dataset (further explanation below). The *psb*A marker was amplified using primers psbAF1 paired with either psbAR2 or, in some cases, psbAR1 [46]. Amplicons were sequenced using the appropriate reverse primer. The *rbc*L marker was amplified in two overlapping pieces using primers F57

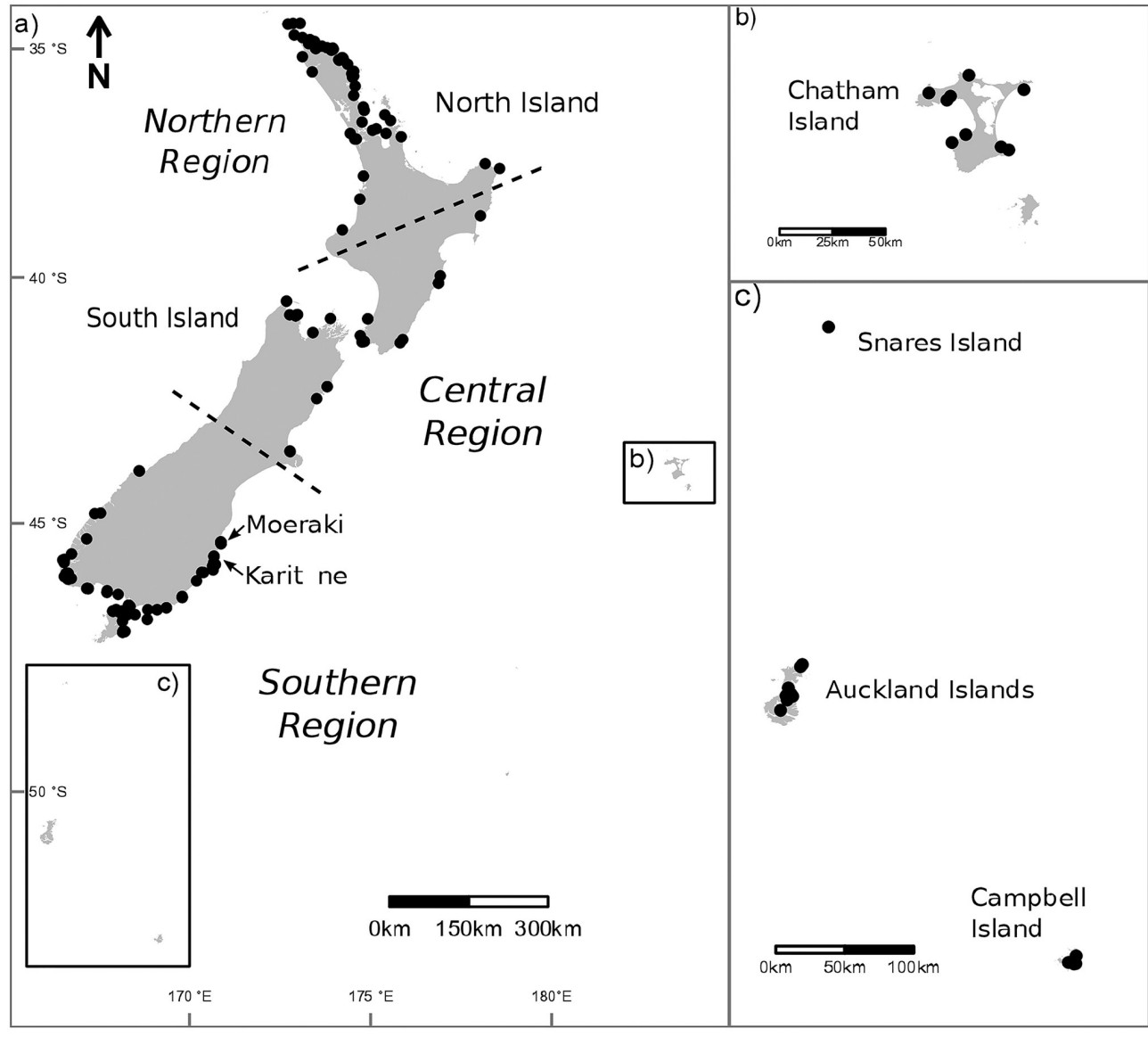

**Fig 1. Collection sites of coralline algae around the New Zealand coast and associated islands.** The dashed lines represent divisions between the Southern, Central and Northern collection regions. Map was created using QGIS software [45] with the use of New Zealand Transverse Mercator 2000 (NZTM2000) projection (LINZ; Land Information New Zealand).

paired with R1150, and F753 with RrbcS [47]. Products were sequenced in both directions using the same primers.

Each PCR reaction contained 3 μl of 1:100 diluted coralline algae DNA extract, 25 pM of each primer, 5 nM of dNTPs, 1x reaction buffer (containing $MgCl_2$ to a final concentration of 1.5 mM), and 0.5 U Kapa 2G Robust HotStart DNA polymerase (Sigma-Aldrich, St Louis, MO). Amplification conditions were: Initial denaturation of 95˚ C for 3 min; 35 cycles of either 95˚ C for 30 s, 41˚ C for 20 s, 72˚C for 1 min (*psb*A) or 95˚ C for 30 s, 48˚ C for 15 s, 72˚C for 30 s (*rbc*L); followed by a final extension of 2–3 min at 72˚ C. PCR products were checked for size and concentration by electrophoresis in 1% agarose gels, purified using ExoSAP-IT reagent (USB, Cleveland, Ohio, USA) and sequenced by Macrogen Inc. (Seoul, Korea).

Sequences were imported into Geneious 11.0.3 (https://geneious.com), trimmed to remove poor quality sequence and assembled to generate a consensus sequence as necessary.

## Species delimitation and phylogenetic analyses

The *psb*A data was analysed as two separate datasets of taxa belonging to the Corallinophycidae: (1) members of the order Corallinales, and (2) members of the orders Hapalidiales and Sporolithales. Sequence data from the southern New Zealand specimens were aligned with sequence data previously obtained from collections of Corallinophycidae specimens from central and northern regions of New Zealand (see [17,41,42,48]). Species delimitation methods were employed on these two *psb*A datasets to establish primary species hypotheses (PSH). Prior to analysis, identical sequences were removed using a python script as recommended by Blair & Bryson [49]. Three separate single-locus delimitation methods were implemented: (1) a distance based method, Automatic Barcode Gap Discovery (ABGD [50]); (2) a tree-based method, Poisson-Tree Processes (PTP [51]); and (3) an ultrametric tree based method, Generalized Mixed Yule Coalescent (GMYC [52,53]). AGBD was run on the alignments using the web interface (http://wwwabi.snv.jussieu.fr/ public/abgd/) with the Jukes-Cantor (JC69) distance and relative gap width (X) set to 0.5. A Bayesian version of PTP (bPTP) with 500,000 MCMC generations was run with thinning every 100 generations on the web server (http://species.h-its.org/ptp) using a standard maximum likelihood phylogeny as an input. Maximum likelihood (ML) phylogenies were constructed with PhyML v3.1 [54] using the General Time Reversible (GTR) sequence evolution model [55] with rate variation modelled as a discrete gamma distribution [56], and including a parameter for invariant sites (GTR+I+G). This sequence evolution model was selected for both datasets using the Bayesian Information Criterion (BIC) in jModelTest v2.1.10 [57]. The single threshold version of GMYC (sGMYC) was calculated by using an ultrametric tree as an input on the web server (http://species.h-its.org/gmyc). The ultrametric tree was constructed using BEAST v2.4.8 [58] running for 10 million MCMC generations with sampling every 1000 generations. The clock model was set to 'relaxed lognormal molecular clock', and the prior 'coalescence tree with constant population' was chosen [59]. A conservative approach, similar to that employed by Hoshino et. al [60], was used to define the PSH. This was based on a consensus approach where a PSH was considered supported when two or more species delimitation methods supported it. One specimen was selected from each PSH (hereafter referred to as a species) for *rbc*L sequencing.

A concatenated *psb*A and *rbc*L alignment was created for each of the three orders (Corallinales, Hapalidiales and Sporolithales) using one specimen from each identified species. We also included sequences available in Genbank that were obtained from type material, to provide reliable taxonomic reference points for existing names (S1 Table). Each analysis included four taxa from the other two orders, and also *Corallinapetra novaezelandiae* T.J.Farr, W.A.Nelson & J.E.Sutherl. (which is excluded from each of the three currently recognized orders), as outgroups.

Maximum likelihood trees were constructed using IQ-Tree [61] for each of the three orders. Appropriate partitioning strategies and models of sequence evolution were estimated under Bayesian Information Criterion (BIC) using ModelFinder [62] implemented in IQ-Tree (S2 Table). The robustness of internal nodes was assessed by approximate Likelihood Ratio Tests (aLRT) based on Shimodaira-Hasegawa (SH)-like procedures [63] and by 1,000 bootstrap replicates, both implemented in IQ-Tree. MrBayes v3.2 [64] was used for the Bayesian analysis. Not all sequence evolution models available in IQ-Tree can be implemented in MrBayes, therefore ModelFinder in IQ-Tree was re-run on the partitioned dataset to estimate the best sequence evolution models among those available in MrBayes (S2 Table). Two independent

analyses, each with four independent chains were run on the partitioned datasets for 5,000,000 generations, sampling every 1,000 generations. Burn-in was assessed by inspection of average parameter values and log-likelihood plots in Tracer v1.6 [65] and confirmed by inspection of PSRF values calculated in MrBayes. All phylogenetic trees were visualised in FigTree v1.4.3 [66].

## Species richness

Non-parametric incidence-based asymptotic estimators were used to estimate the number of species expected in New Zealand and the three sub regions (Southern, Central and Northern regions) if additional sampling was to take place in similar habitats [67]. Three different estimators were calculated: (1) the Chao2 estimator [68], (2) the first order Jackknife (Jack1), and (3) the second order Jackknife (Jack2). To visualise the relationship between number of samples and the number of species discovered, a species accumulation curve (SAC) and 95% confidence interval was constructed using a permutational approach (permutations = 999). Incidence-based estimators were calculated using the '*specpool*' function and the SAC using the '*specaccum*' function in the R package 'vegan' [69].

# Results

## Species delimitation

A total of 122 species were identified in southern New Zealand coralline algal samples using species delimitation methods (ABGD, sGMYC and bPTP) based on *psb*A sequence data (S3 Table). Of these, 57 were identified in the order Corallinales (Fig 2 and S3 Table), 61 in the order Hapalidiales (Fig 3 and S3 Table), and four in the order Sporolithales (Fig 3 and S3 Table). There was broad agreement among the three species delimitation methods, although there were some differences in the species identified. A total of 18 singleton species (i.e. species known from a single collection) are represented in southern New Zealand and 29 New Zealand-wide (S3 Table).

## Phylogenetic results

Sporolithales: Four species within the order Sporolithales were identified in the New Zealand region (Fig 4 and S4 Table). These included two species of *Sporolithon* Heydr., one *Heydrichia* R.A.Towns., Y.M.Chamb. & Keats, and one taxon, Sporolithales sp. A NZC2014, which does not resolve clearly within either *Sporolithon* or *Heydrichia*. Taxa from this order (Sporolithales) were predominantly found in central and northern regions of New Zealand reported in Nelson et al. [17]. *Sporolithon* sp. B NZC2374 has been recorded only as rhodoliths, while all records of *Sporolithon* sp. A NZC2175 are epilithic.

Corallinales: Our study identifies 59 Corallinales species in the New Zealand region (Fig 5 and S4 Table). Two taxa from the northern Kermadec Island are identified as *Neogoniolithon* sp. and *Mastophora pacifica* (Heydr.) Foslie, although this identification has not been verified by sequence data from the type specimen of *M. pacifica* and remains provisional. The remaining 57 species are found in the main part of the New Zealand archipelago. Twelve are resolved in the Corallinaceae, including eight species of *Jania* J.V.Lamour., only two of which can be confidently assigned species names (*Jania sagittata* (J.V.Lamour.) Blainv. and the recently described *Jania sphaeroramosa* Twist, J.E.Sutherl. & W.A.Nelson), three species of *Arthrocardia* J.V.Lamour. and one species of *Corallina* L. formerly assigned to *C. caespitosa* R.H.Walker, J.Brodie & L.M.Irvine. This taxon shows remarkable morphological variability, and some (1–2 bp) genetic variability in *psb*A sequence data.

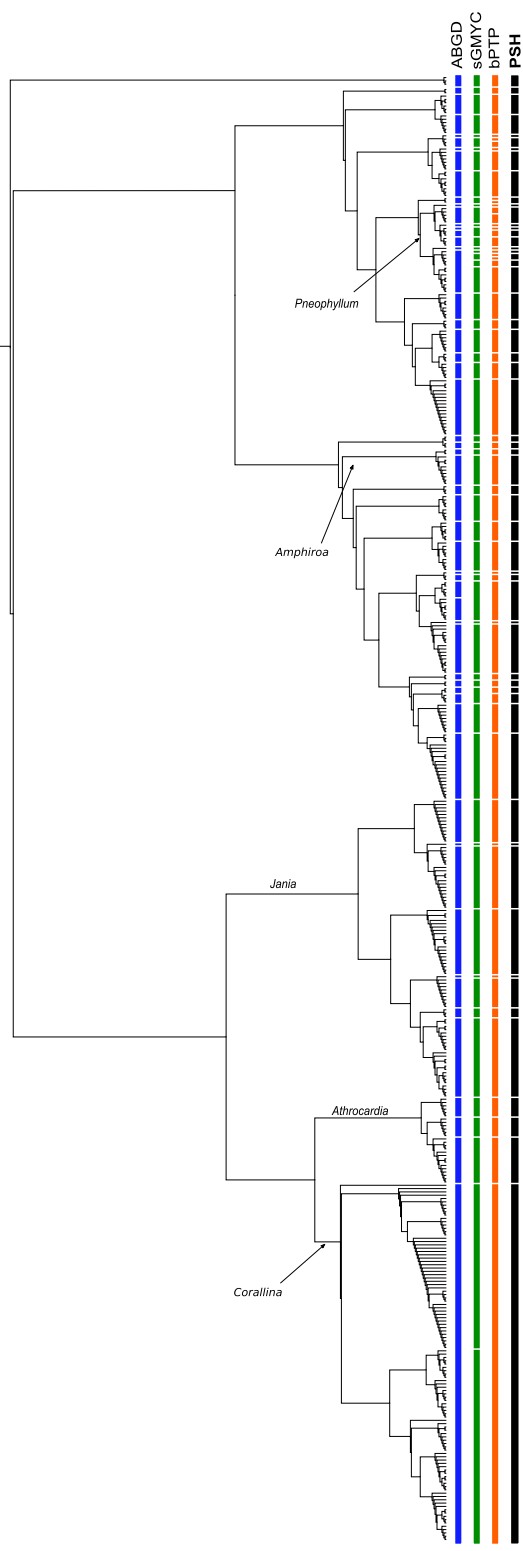

**Fig 2. Maximum likelihood *psb*A tree for the order Corallinales showing species delimitation methods.**
ABGD = Automatic Barcode Gap Discovery, sGMYC = single threshold Generalized Mixed Yule Coalescent,
bPTP = Bayesian Poisson-Tree Processes, and PSH = Primary Species Hypothesis—assigned using a consensus
approach where two or more species delimitation methods agreed.

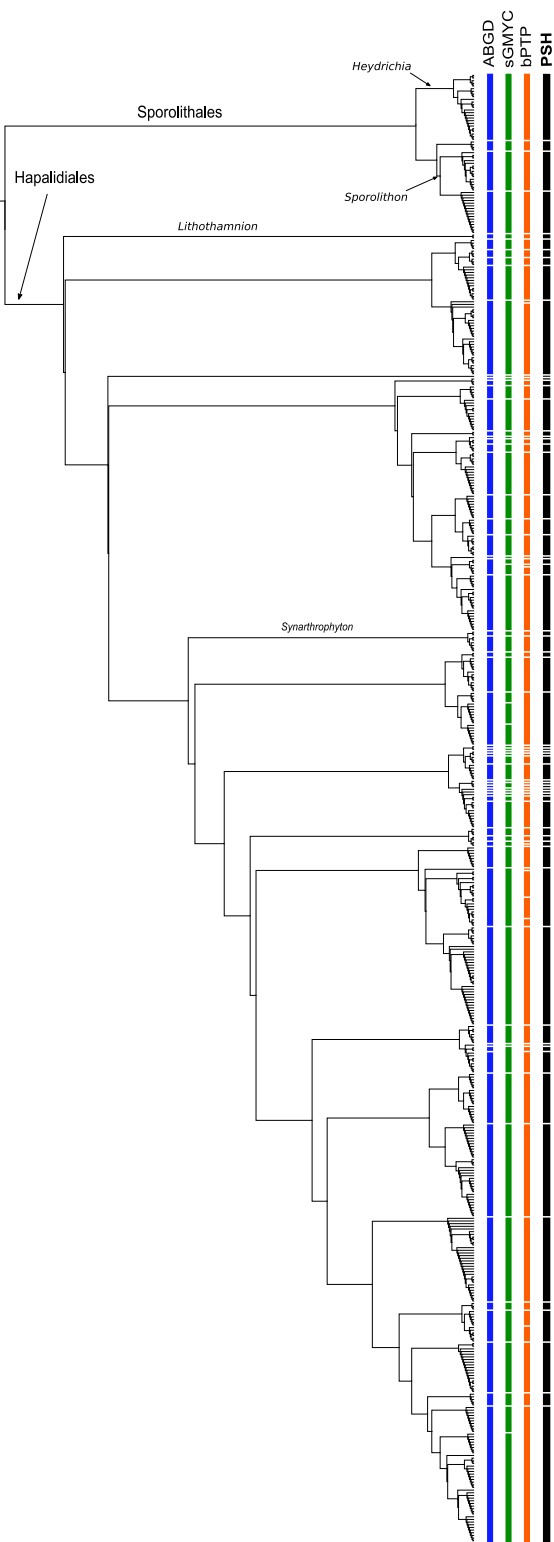

**Fig 3. Maximum likelihood *psb*A tree for the order Hapalidiales and Sporolithales showing species delimitation methods.** ABGD = Automatic Barcode Gap Discovery, sGMYC = single threshold Generalized Mixed Yule Coalescent, bPTP = Bayesian Poisson-Tree Processes, and PSH = Primary Species Hypothesis—assigned using a consensus approach where two or more species delimitation methods agreed.

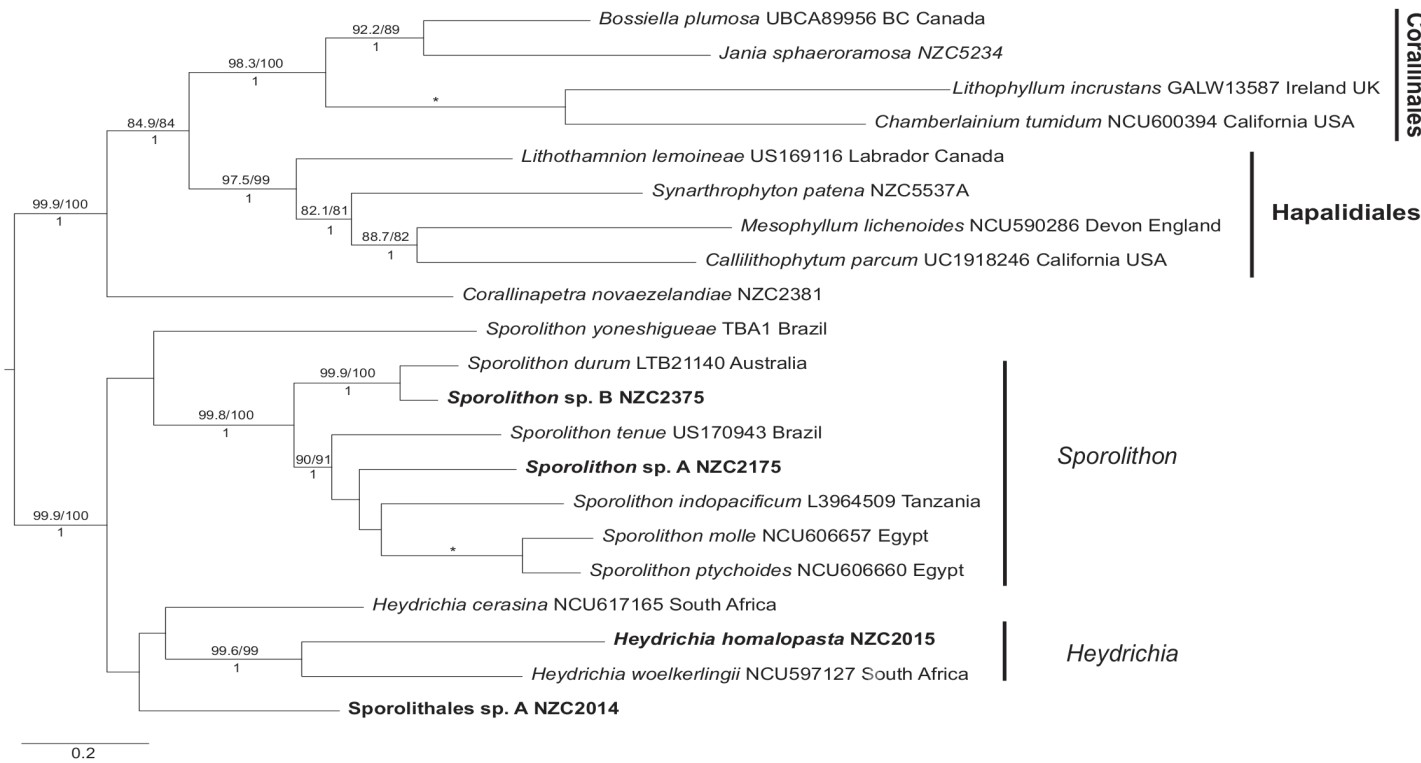

**Fig 4. Maximum likelihood (ML) phylogeny of concatenated *psb*A and *rbc*L alignment with members of the order Sporolithales, showing the relationships between New Zealand (in boldface) and global taxa.** Genera with two or more species which include a New Zealand representative are indicated by a vertical line and a genus label. Approximate Likelihood Ratio Test (aLRT) values (%) followed by ML bootstrap values (%) are shown above each branch and Bayesian Posterior Probabilities (PP) are shown below. Support values are shown if two of the three values are greater than 80%. An asterisk represents support at 100/100/1.

Of the 45 remaining Corallinales species, only one can be assigned to a described species, the geniculate *Amphiroa anceps* (Lam.) Decne. Eleven taxa can be assigned to genus *Pneophyllum* Kütz. on the basis of their resolution with sequence data from the type specimen of *Pneophyllum limitatum* (Foslie) Y.M.Chamb. Sixteen species are resolved within the Lithophylloideae but do not resolve with members of either *Lithophyllum* Phil. or *Amphiroa* J.V. Lamour., despite a number of these being initially assigned as *Lithophyllum* spp. based on a select number of morpho-anatomical features. The remaining 17 taxa are not resolved with type material from any genus or species within Corallinales. On the tree we have marked six well supported clades that may represent new genera with multiple members within the New Zealand region. These are annotated as 'Genus N' where N is an integer. Numbers for these hypothetical genera follow Twist [70]. Outside of these clades ten specimens are not closely resolved with any other taxa in the analysis (Fig 5).

Hapalidiales: Sixty one species were identified within Hapalidiales (Fig 6 and S4 Table). Of these, only two can be identified to genus and species: *Lithothamnion crispatum* Hauck, which is resolved with strong support with other *Lithothamnion* species, and *Synarthrophyton patena* (Hook.f. & Harv.) R.A.Towns., which is derived from material from the type locality, and conforms to the description of that species. Based on currently available information, the remaining 59 species cannot be assigned to any genus or species within the order. For instance, none of the New Zealand taxa are resolved with the type species of *Mesophyllum*, *M. lichenoides*

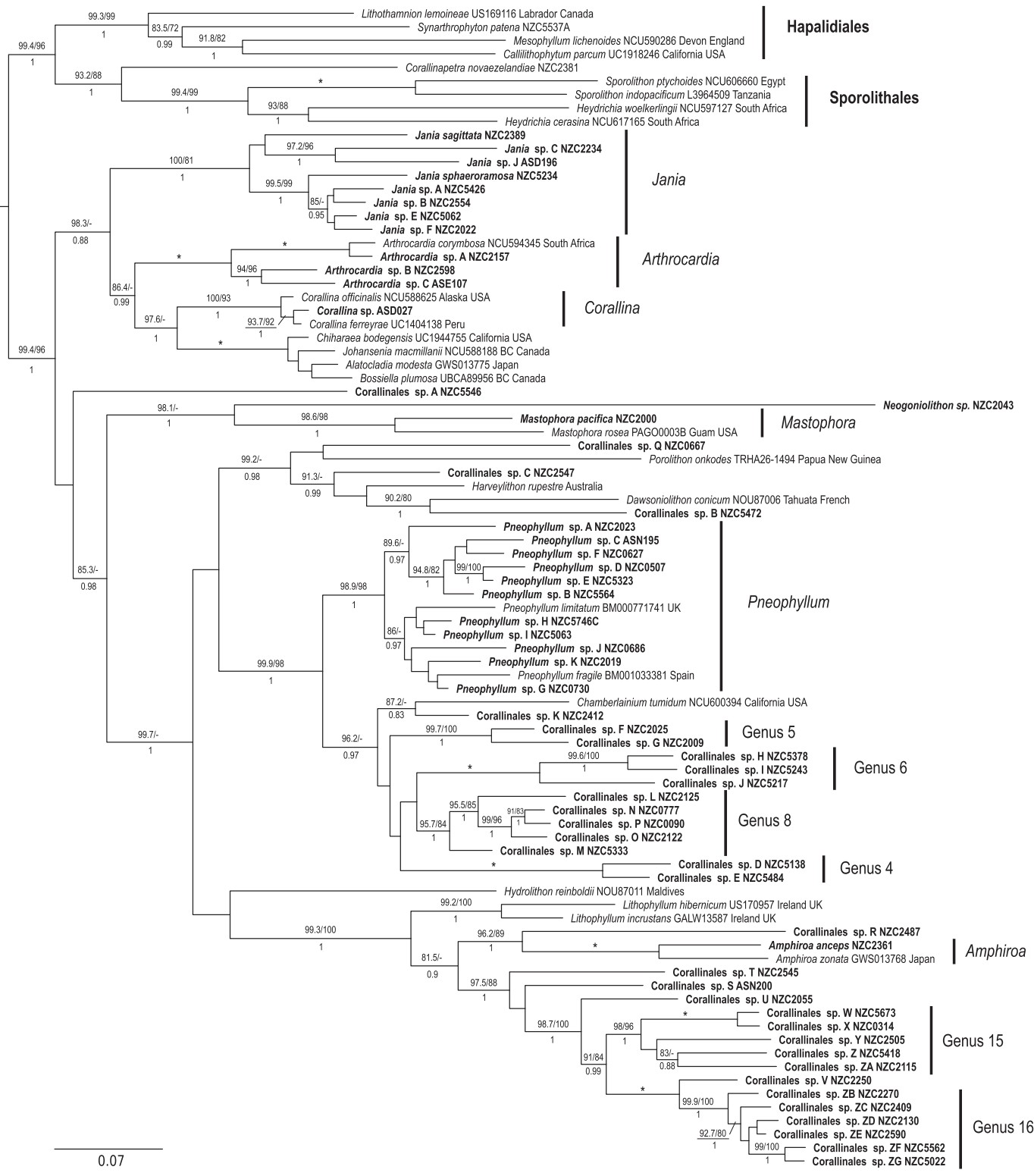

**Fig 5. Maximum likelihood (ML) phylogeny of concatenated *psb*A and *rbc*L alignment with members of the order Corallinales, showing the relationships between New Zealand (in boldface) and global taxa.** Hypothetical genera are indicated for well supported clades containing closely related species within the New Zealand dataset. Genera with two or more species which include a New Zealand representative are indicated by a vertical line and a genus label. Approximate Likelihood Ratio Test (aLRT) values (%) followed by ML bootstrap values (%) are shown above each branch and Bayesian Posterior Probabilities (PP) are shown below. Support values are shown if two of the three values are greater than 80%. An asterisk represents support at 100/100/1.

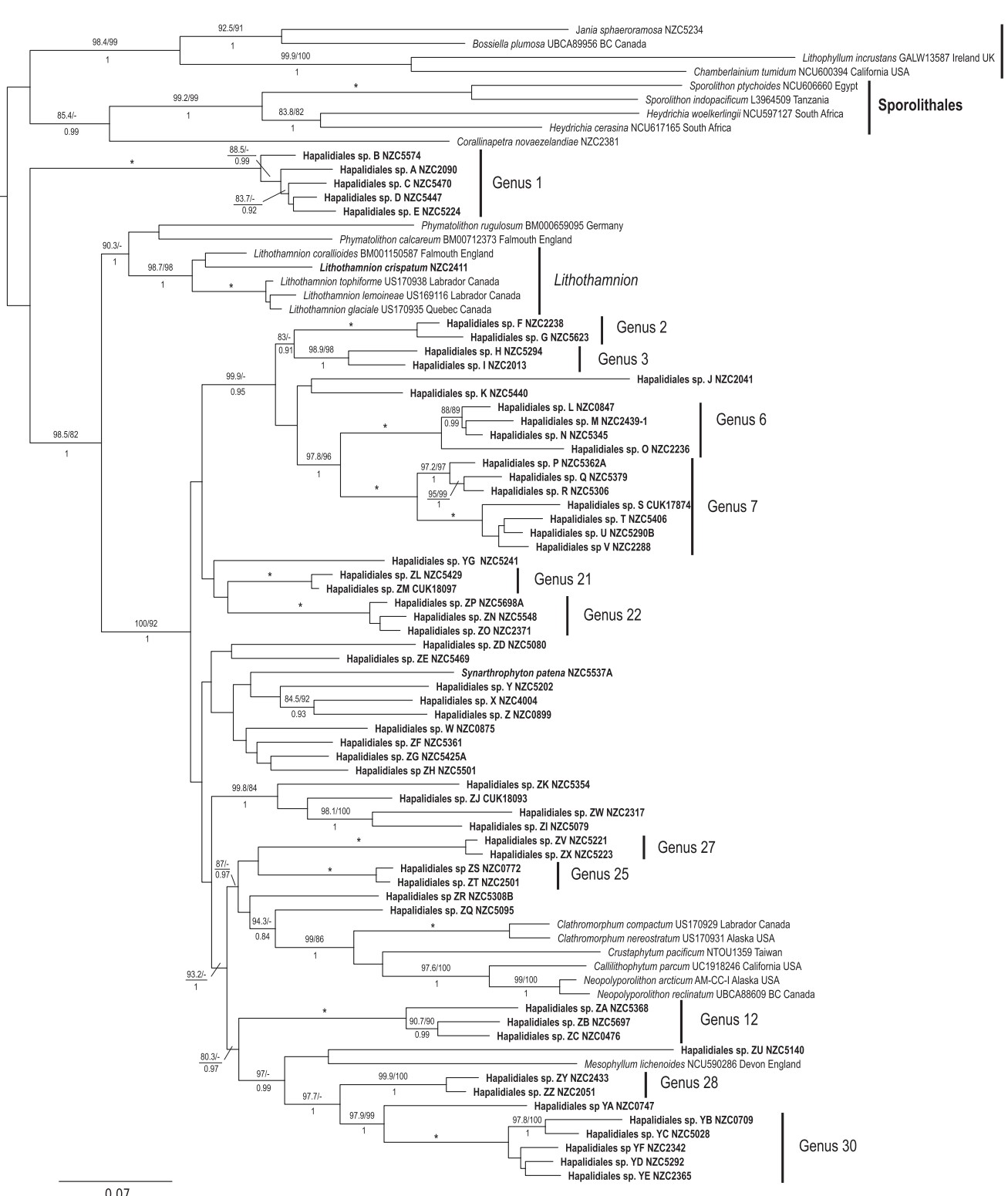

**Fig 6. Maximum likelihood (ML) phylogeny of concatenated *psb*A and *rbc*L alignment with members of the order Hapalidiales, showing the relationships between New Zealand (in boldface) and global taxa.** Genera with two or more species which include a New Zealand representative are indicated by a vertical line and a genus label. Approximate Likelihood Ratio Test (aLRT) values (%) followed by ML bootstrap values (%) are shown above each branch and Bayesian Posterior Probabilities (PP) are shown below. Support values are shown if two of the three values are greater than 80%. An asterisk represents support at 100/100/1. Hypothetical genera are based on well supported clades containing closely related species within the New Zealand dataset.

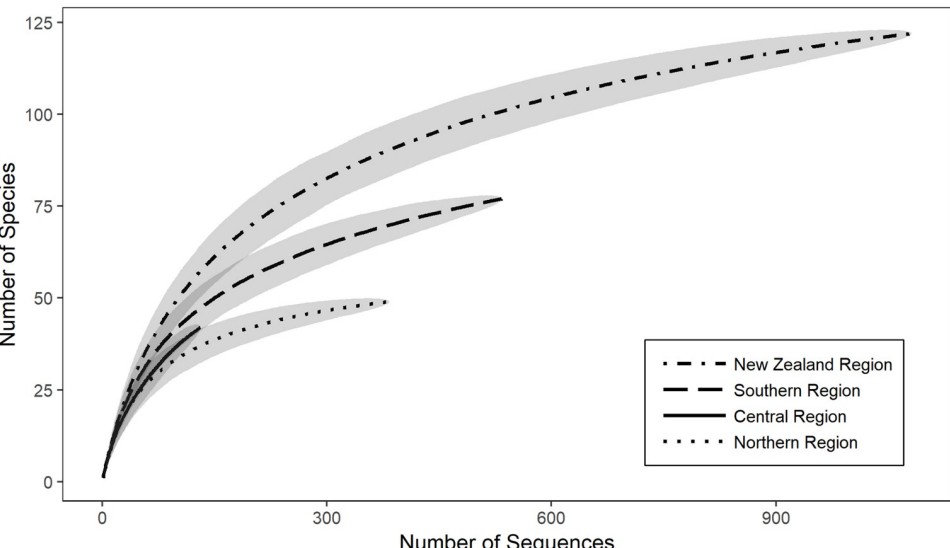

**Fig 7. Species accumulation curve for coralline algal sequences from southern, central and northern New Zealand, and for all of New Zealand.** Smoothed curve with 95% confidence interval (grey area) calculated using a permutational approach (n = 999). Note that this plot shows the accumulation of species recorded in each region and does not extrapolate to predict the total number of species.

(J.Ellis) Me.Lemoine, despite initially being assigned to this genus based on morpho-anatomical features, and consequently none can be assigned to *Mesophyllum*.

Within the New Zealand Hapalidiales twelve well supported lineages are observed. These are indicated on the Fig as 'Genus N' where N is an integer from 1 to 30. Outside of these 12 lineages and the two named taxa, 20 sequences are resolved without any close relationships in the New Zealand Hapalidiales dataset.

## Species diversity

Species accumulation curves indicate that the discovery of new species in all three New Zealand regions have not reached an asymptote, and a significant proportion of species diversity remains undiscovered (Fig 7). Estimates of species diversity of coralline algae predicted in New Zealand if sampling were to continue range from 141.7 ± 9.27 for the Chao2 estimator (Table 1), 150.9 ± 5.3 for Jack1 and 157.9 for Jack2. This high level of undiscovered diversity is further supported by the observation that 29 species are represented by only a single sequence (singleton) in the New Zealand wide dataset.

**Table 1. Number of coralline algal species from the three orders (Corallinales, Hapalidiales and Sporolithales) estimated using Chao2 incidence-based species estimators (see text for SD of the estimates), rounded down to the nearest integer, from each study regions around the New Zealand coast.** The number of species (distinguished by PSH) identified in this study by species delimitation methods are recorded in brackets. n = the number of sequences used in analyses.

|  | Southern (n = 535) | Central (n = 130) | Northern (n = 384) | NZ wide (n = 1049) |
|---|---|---|---|---|
| Corallinales | 44 (29) | 30 (18) | 31 (29) | 62 (57) |
| Hapalidiales | 54 (47) | 27 (21) | 18 (16) | 75 (61) |
| Sporolithales | (1) * | 3 (3) | 4 (4) | 4 (4) |
| Total | 99 (77) | 60 (42) | 53 (49) | 141 (122) |

*No incidence-based species estimator could be calculated due to only one individual being found

The southern New Zealand region is predicted to have the highest number of coralline algal taxa with estimates of 99 ± 12.4 for the Chao2 estimator (Table 1), 98.9 ± 4.7 for Jack1 and 109.9 for Jack2. This is followed by the central region with estimates of 60 ± 10.9 for the Chao2 estimator, 60.9 ± 4.3 for Jack1 and 69.8 for Jack2, and finally by the northern region with estimates ranging from 53.5 ± 4 for the Chao2 estimator, 57.9 ± 3 for Jack1 and 58 for Jack2. The differences between the numbers of coralline algae predicted in each region are largely driven by the higher number of Hapalidiales taxa predicted to be present in the southern region compared to central and northern regions (54 species predicted compared to 27 and 18, respectively).

## Species diversity at small spatial scales

A total of 17 species were identified at both Butterfly Bay and Moeraki. If futher sampling was to occur an additional 16 (total of 33) and eight (total of 25) species respectively are predicted using Chao2 estimates. At Moeraki, nine out of the 17 species had average abundance ≤3% cover with only two species having >18% average cover (S5 Table).

## Discussion

The focus of this study was to document coralline algal diversity in southern New Zealand, and to compare this with diversity in other parts of the New Zealand region, using recent phylogenetic studies of coralline algae (Corallinophycidae) as the framework for the analyses. We anticipated that there would be a number of new discoveries using a collection strategy that targeted a range of habitat types across a wide geographic area and employing molecular systematics, as well as the application of techniques to predict species diversity. However, the number of genera and species of coralline algae that were distinguished amongst the collections (122 species), and the diversity that was predicted (141 species), exceeded both our expectations, and earlier estimations based on morpho-anatomical approaches [71]. In addition, the discovery of high diversity at small spatial scales (not apparent from morphological features or readily distinguished in the field) has significant implications for the design of future sampling, and interpretation of field data.

### Species diversity

The confirmation of high species diversity in New Zealand is consistent with recent studies in tropical coral reefs [24], rhodolith beds in the United Kingdom [72], geniculate species around the South African coast [73], corallines in the Mediterranean Sea [74,75], and around the Brazilian coast [25]. It has been predicted that global diversity in coralline algae is likely to be 2–4 times higher than previously assigned using morpho-anatomical approaches [20,26,76]. Contrary to most other organism groups, diversity of macroalage has been shown to peak at mid to high latitudes, with several competing hypotheses proposed related to stability of enviromental variables, biotic interactions and oceanic currents [77–79], which may explain the high regional diversity observed in this study across the New Zealand region spanning from mid latitudes of 29° S to high latitudes of 52° S.

What distinguishes the diversity reported here is the number of taxa which we were unable to assign to genus and species: there are 49 genera and 115 species that are distinct from any authoritatively defined species for which sequence data are currently available, with further diversity remaining to be discovered. While the New Zealand region is known for high species diversity and endemism, a consequence of both the geographic extent of the region, from the subtropical Kermadec Islands (29°S) to subantarctic Campbell Island (53.5°S), and the geological history of continental Zealandia and isolation from other landmasses [80], these discoveries at small local and regional scales exceed diversity reported elsewhere and also raise many

questions. Almost a quarter of the species discovered to date are 'singleton' species, i.e., known from single collections (29 out of 122 species in the New Zealand region). Without further collections it is not possible to even begin to understand the ecological or geographic distributions of these 'singleton' species nor to characterise them morphologically or anatomically.

It is not clear how many of the currently undescribed genera in the New Zealand region will prove to be restricted to this region. Hind et al. [28] reported the NE Pacific "to be a center of endemism for both geniculate and nongeniculate coralline genera" with four endemic geniculate genera and two endemic nongeniculate genera. Based on currently available information, very few New Zealand species are found in neighbouring regions or globally. Where we have multiple collections of a species it has been possible to examine geographic ranges, and in such cases, we have found distributions of individual species vary from a few that are relatively widespread in the New Zealand region to other species that appear to have highly restricted distributions.

### Consequences of high diversity

Diversity is important for maintaining a range of ecosystem functions (e.g. habitat provisioning, facilitation of larval settlement, and carbon storage) and loss of diversity can result in major losses and changes to functionality [81,82]. Loss of diversity has the potential to drive ecosystem changes comparable to those induced by environmental stressors such as climate warming or increased ocean acidification [81]. Not only common or dominant species are important: rare species have been shown to be more susceptible to temporal variance in disturbances than abundant species, and, because of this, are important in driving changes in community structure and assemblage dynamics under predicted climate change [83]. Given the ubiquity of coralline algae in euphotic marine environments [e.g. 84] and the foundational roles they play, losses or declines in coralline species could have significant impacts on the ecosystems in which they are found.

Recent interest in ocean acidification and climate change has resulted in a many studies examining their impacts on coralline algae (e.g. [4,5,38,85,86]). The need for appropriate experimental design and methodology to adequately investigate the interactions and cumulative impacts of key stressors on coralline algae has been highlighted, with guidance given on best practices (e.g. [5,87]). However, there is also a critical need to base such research on authoritatively identified material, that allows for reliable, reprodicible comparisons between species, locations and studies. It is critical that material has been identified to reflect the most up-to-date taxonomic understanding, and that representative voucher material is deposited in a publicly accessible herbarium for future examination, for example when taxonomic hypotheses change. For example, Gabrielson et al. [24] revealed that the name *Porolithon onkodes* (Heydr.) Foslie–the most widely reported reef-building coralline algal species–has been applied to more than 20 species, and Pezzolesi et al. [30] reported more than 13 species within the *Lithophyllum stictiforme* (Aresch.) Hauck complex in Mediterranean coralligenous habitats. With increasing recognition that the extent and impact of environmental stressors can be species-specific (e.g. [8,38,85,86]), and that coralline algae can differ in their ecological traits (e.g. growth and competitive ability [11,33–35,88]), in addition to ecological functions they provide (e.g. influence on algal and invertebrate recruitment [9–11,89]), it is clear that taxonomic confusion can compromise otherwise excellent science and lead to false conclusions. The importance of species recognition in order to understand the community processes is demonstrated in the study of Hind et al. [90] who found that although coralline algae are more abundant in urchin 'barrens', these communities are dominated by only a few species and have lower diversity compared to coralline assemblages under intact kelp forests. This work by

Hind et al. [90] is a good example of a study in which enhanced resolution and understanding of nearshore community dynamics was dependent upon reliable species identifications using molecular data.

### Consequences for taxonomy of New Zealand coralline algae

This study indicates that many names previously applied to New Zealand coralline algae (e.g. [17,41,42,48]) were incorrect. This was a consequence of several issues—placing weight on morpho-anatomical characters that are now understood to be insufficiently informative for species recognition, poorly understood generic and specific boundaries, and the lack of comparative molecular data globally (particularly data from type material) which made placement in a global context difficult in these earlier studies. For example, *Spongites* Kütz. and *Pneophyllum* had been distinguished from each other by the mode of their tetra/bisporangial conceptacle roof development, and, in Australia and New Zealand, the substratum type was used to assign specimens to either *Pneophyllum* (epiphytic) or *Spongites* (epilithic, epizoic, or unattached) [91]. According to Caragnano et al. [19], this assignment by habitat led to misidentifications and to the polyphyletic outcomes seen in DNA sequence-based analyses.

The coralline algal distributions revealed in this study depart from the pattern of diversity seen across the rest of the New Zealand macroalgal flora, where the greatest diversity has been recorded in the northern North Island [92]. In the Corallinophycidae, the order Hapalidiales is particularly well represented in southern New Zealand with 47 species (54 predicted) present, compared to 21 species (27 predicted) and 16 species (18 predicted) in the central and northern areas, respectively. High diversity in Hapalidiales is also known to occur in the northern Atlantic [31] in the boreal–subarctic transition zone that spans the North Atlantic from North America (Gulf of Maine and the southern Canadian Maritimes) to southwestern Iceland and the Norwegian outer coast between 42-63˚ N. In contrast, the Sporolithales in New Zealand is better represented in the north (three genera, four species) than the south (one species). This pattern is also seen globally where the majority of members of this order are found in tropical to warm temperate regions [93]. The Corallinales is represented by similar numbers of taxa in northern and southern New Zealand although the estimators predict that more will be found in the south. In their analyses of the Corallinales, Rösler et al. [94] reported a clade of southern hemisphere taxa (from New Zealand, Southern Australia, South Africa, and Chile) which they considered warranted further investigation. They also commented on subclades of specimens from New Zealand observed which they considered "indicates a high degree of genetic differentiation of corallines from this region".

### Conclusions

The recognition of coralline diversity has significant implications for conducting future investigations and interpreting past research. To ensure reproducible science an understanding of diversity within a molecular phylogenetic framework is critical for progress in all areas where coralline algae are the subject–ecology, physiology, responses to global changes, calcification, cell wall materials. If voucher material has been retained and deposited in publicly accessible herbaria or collections, and sequence data are available, the identity of species used for experiments may later be confirmed in the light of new knowledge and emerging understanding of coralline algal diversity. Without such voucher material, there are significant challenges in relating newly discovered diversity to earlier published accounts: the identity of species used for experiments cannot be confirmed, and thus conclusions regarding species specific ecosystem services and responses to disturbances cannot be validated.

While this research has revealed high diversity of coralline algae in the New Zealand region of the south west Pacific, further research involving targeted collection programs, multigene phylogenetic analyses and morpho-anatomical characterisation, is needed before relationships and diversity of the New Zealand flora can be fully understood. In addition, clarification of the type material for New Zealand species is required [71]. To enable the New Zealand flora to be placed in a wider context, and to understand phylogeographic relationships, there is a need for detailed investigations of the coralline floras of other regions (particularly in the southern hemisphere) focused on documenting diversity, increasing taxon sampling and distributional data. Given that different species are likely to exhibit different ecological traits, perform different functions, and respond differently to stressors, it is paramount to use molecular methods and to continually develop the taxonomy and systematics of coralline algae to allow for reliable identifications in order to advance future research and fully understand their role in coastal ecosystems.

## Supporting information

**S1 Table. List of global coralline algal type specimens sourced from GenBank for use in concatenated *psb*A and *rbc*L phylogenetic analyses.** Specimens are listed alphabetically within the three orders.— = no data available.
(PDF)

**S2 Table. Partition schemes and sequence evolution models used for combined *psb*A and *rbc*L datasets in implementing (A) maximum likelihood phylogenetic analysis and (B) Bayesian phylogenetic analysis.**
(PDF)

**S3 Table. Collection data for coralline algae samples successfully amplified with *psb*A gene from the New Zealand region.** Species is the phylogenetically derived primary species hypothesis (PSH) the specimen (identified by unique algae number) belongs to. Depth is the metres below mean low water.
(PDF)

**S4 Table. List of representative New Zealand coralline algal species, identified using species delimitation methods, used in concatenated *psb*A and *rbc*L phylogenetic analyses.** Hypothetical genera were based on well supported clades containing closely species within the New Zealand dataset. Specimens are listed alphabetically within the three orders.— = no data available.
(PDF)

**S5 Table. The average percentage cover and relative abundance ± 1 Standard Error (SE) of coralline algal species across a series of six boulders in Moeraki, Otago, New Zealand.**
(PDF)

## Acknowledgments

We would like to thank a large number of individuals (too many to list here), for their assistance in the field and with collections of material. In particular we would like to thank Tracy Farr for her work on corallines in central and northern New Zealand, and the support we have received from the University of Otago, in particular Chris Hepburn, as well as from Ngāi Tahu (Ngāi Tahu Te Tiaki Mahinga Kai).

## Author Contributions

**Conceptualization:** Brenton A. Twist, Kate F. Neill, Judy E. Sutherland, Wendy A. Nelson.

**Data curation:** Brenton A. Twist, Kate F. Neill.

**Formal analysis:** Brenton A. Twist, Judy E. Sutherland.

**Funding acquisition:** Wendy A. Nelson.

**Investigation:** Brenton A. Twist, Kate F. Neill, Jaret Bilewitch, So Young Jeong, Judy E. Sutherland, Wendy A. Nelson.

**Methodology:** Brenton A. Twist, Judy E. Sutherland, Wendy A. Nelson.

**Project administration:** Brenton A. Twist, Wendy A. Nelson.

**Supervision:** Judy E. Sutherland, Wendy A. Nelson.

**Validation:** Brenton A. Twist, Kate F. Neill, Judy E. Sutherland, Wendy A. Nelson.

**Visualization:** Brenton A. Twist, Judy E. Sutherland.

**Writing – original draft:** Brenton A. Twist, Wendy A. Nelson.

**Writing – review & editing:** Brenton A. Twist, Kate F. Neill, Jaret Bilewitch, So Young Jeong, Judy E. Sutherland, Wendy A. Nelson.

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
