## [Decision Letter · Decision Letter 0]

3 Oct 2019

PONE-D-19-23390

High diversity of coralline algae in New Zealand revealed: knowledge gaps and implications for research on the roles of these foundation species

PLOS ONE

Dear Dr. Twist,

Thank you for submitting your manuscript to PLOS ONE. After careful consideration, we feel that it has merit but does not fully meet PLOS ONE’s publication criteria as it currently stands. Therefore, we invite you to submit a revised version of the manuscript that addresses the points raised during the review process.

We would appreciate receiving your revised manuscript by Nov 17 2019 11:59PM. To enhance the reproducibility of your results, we recommend that if applicable you deposit your laboratory protocols in protocols.io, where a protocol can be assigned its own identifier (DOI) such that it can be cited independently in the future. For instructions see: http://journals.plos.org/plosone/s/submission-guidelines#loc-laboratory-protocols

We look forward to receiving your revised manuscript.

Kind regards,

Tzen-Yuh Chiang

Academic Editor

PLOS ONE

**Journal Requirements:**

3. We note that  Figure(s) 1 in your submission contain [map/satellite] images which may be copyrighted. All PLOS content is published under the Creative Commons Attribution License (CC BY 4.0), which means that the manuscript, images, and Supporting Information files will be freely available online, and any third party is permitted to access, download, copy, distribute, and use these materials in any way, even commercially, with proper attribution. For these reasons, we cannot publish previously copyrighted maps or satellite images created using proprietary data, such as Google software (Google Maps, Street View, and Earth). For more information, see our copyright guidelines: http://journals.plos.org/plosone/s/licenses-and-copyright.

a) You may seek permission from the original copyright holder of Figure(s) [#] to publish the content specifically under the CC BY 4.0 license.  

Additional Editor Comments (if provided):

Reviewers' comments:

Reviewer's Responses to Questions

**Comments to the Author**

1. Is the manuscript technically sound, and do the data support the conclusions?

Reviewer #1: Yes

Reviewer #2: Yes

2. Has the statistical analysis been performed appropriately and rigorously? 

Reviewer #1: Yes

Reviewer #2: N/A

3. Have the authors made all data underlying the findings in their manuscript fully available?

Reviewer #1: Yes

Reviewer #2: Yes

4. Is the manuscript presented in an intelligible fashion and written in standard English?

Reviewer #1: Yes

Reviewer #2: Yes

5. Review Comments to the Author

Reviewer #1: The manuscript entitled High diversity of coralline algae in New Zealand revealed: knowledge gaps….by Brenton Twist and co-authors is an important original contribution to the knowledge of the biodiversity of a globally distributed group of habitat engineers, the coralline algae. The paper provides the results of a country-scale investigation of coralline diversity, based exclusively on molecular genetics, and a statistical estimate of the still unknown coralline species in the studied area, based on the number of new species/specimens identified in the contribution. The paper is very interesting because it is aimed to show the dimension of the unknown or undescribed biodiversity, rather than trying to identify species, although the theme of morphological vs genetic vs integrate taxonomy is also tackled. Therefore the publication is expected to become important not only to the super-specialized readers, but also to a wider audience. For these reasons, I recommend the publication of this paper, after fixing some minor issues that I’m listing below:

Title: I would shorten it by cutting the words on the roles of these foundation species because their role is already assesses elsewhere and the paper does not contribute to the topic (systematics, not ecology)

Line 67: Corallines are not only stabilise coral reef structures but also are the major framework builders in the algal reefs of the temperate latitudes. This must clearly stated in the sentence, along with appropriate reference (i.e. Ballesteros 2006, Bracchi et al 2017 Continental Shelf Research 144:10-20; Bracchi et al. 2019)

Line 67: in the recovery (swap of words)

Line 78: a list of dozens of citations is far too much. I suggest shorten it to a smaller group of references by selecting examples of the contribution of DIFFERENT groups of authors and diverse geographic areas wordwide. Please add reference to Kato works in Japan (for example Kato et al 2011 J of Phycology) and Basso work in the Red Sea/Indian Ocean (Basso et al 2015 Phytotaxa).

Line 93: Please add reference to the revision of L. kaiseri (Basso et al 2015 Phytotaxa)

Lines 173 and following: Please declare which is the higher ranks taxonomic framework for your work. Hapalidiales, Sporolithales etc are used but no reference to high rank taxonomy is provided, and this also has been changing recently, therefore it is needed to specify.

Lines 265-269. These lines are intriguing but unclear. Has something to do with Corallinapetra (ref 16?)? Why do not declaring explicitly? I suggest expanding this in the discussion

Line 396. Not apparent from morphological features. This is the most important issue in the manuscript. The paper does not discuss the morphological characters at all. 1) They are considered as a whole not useful a priori; 2) No coralline identification has never been possible in the field; 3) Badly conducted morphological analyses, such those relying on few badly oriented SEM pictures, will never provide any result. New morphological characters could be found if a serious morphological approach would be performed along with the genetics, which is the approach that I would recommend to follow and that I suggest to declare as recommended in the conclusions of this paper.

Line 431 The chapter title is too long

Line 467 Understanding the nearshore community is not an issue tackled here, please remove.

Line 472 The reference 16 is used in a positive sense previously (see Lines 265-269), while here it is said that identification is incorrect. I understand that this probably referes to some further work in preparation, but you have to try to be a little bit clearer.

Line 512 Please refer here explicitly to the need of further detailed morphological description based on groups emerging from molecular taxonomy.

Reviewer #2: General comment

The manuscript entitled "High diversity of coralline algae in NWZ revealed knowledge gaps and implications for research on the roles of these foundation species" is a thorough study of coralline diversity in NZ conducted by combining an impressive and therefore comprehensive sampling strategy and state of the arts tools of molecular assisted taxonomy. In my mind this study is very well designed and should be published. I have a few suggestions for improvements that are listed in the detail comments.

Detail comments:

l.003 Remove the ':" in the title

l.074 in the list of new orders the reference to the Sporolithales is missing. Le Gall, L., Payri, C.E., Bittner, C.E., & Saunders, G.W. (2010). Multigene polygenetic analyses support recognition of the Sporolithales, ord. nov. Molecular Phylogenetics and Evolution 54(1): 302-305.

l.101 "Although this is a regional study..." it sounds like the authors try to justify themself... I think that the number of species they uncovered is even more striking giving the regional scale!

l.207 The inclusion of type only sequence from GenBank is both a good and not so good idea in my mind. In a taxonomic point of view, I fully understand this strategy; however, in a evolutionary point of view, a phylogeny should include a balanced sampling of all the taxa in the studied group, no matter where they are from... I know that there are a wealth of coralline sequences in GenBank with some rather poorly annotated but I still think that it would provide valuable information to densify a bit the taxa sampling for the phylogenetic inferences of this study. An intermediate strategy may consist in including specimens for which both psbA and rbcL are available... In addition of improving the phylogenies this strategy would allow to assess the distribution of the taxa encountered in New Zealand (are they endemic or not?)

l.213 In my previous experience, with short markers as psbA and rbcL, I tend to find that the partition by codon only (without spliting the dataset by gene) is the best partition. Did this option was investigated with a program such as partition finder?

l.297 It may worth to mention whether the 16 species look like Lithophyllum or Amphiroa based on gross morphology...

l.322 Same here, it may worth to mention if in the field they were identify as putative Mesophyllum...

l.347 I am note sure that it makes sense to have decimal for the estimation of species diversity.

l 399 In this paragraph of discussion dedicated to "species diversity", it would be a good addition to refer to the global distribution of seaweeds such as Keith et al (2014) or Kerswell et al (2006)

Keith S. A., Kerswell A. P. & Connolly S. R. 2014 - Global diversity of marine macroalgae: environmental conditions explain less variation in the tropics: Global diversity of marine macroalgae. Global Ecology and Biogeography 23 (5) : 517–529.

Kerswell A. P. 2006 - Global biodiversity patterns of benthic marine algae. Ecology 87 (10) : 2479–2488.

l.431 This paragraph is excellent!

fig. 2 & fig. 3 With so few annotation, it is very difficult to get oriented in these trees. Would it be possible to add the families or the delimited taxa with a number which refer to a table?

fig. 4 Why no Rhodogorganales are included? This is the sister taxa to the Sporolithales and therefore the best outgroup!

fig. 5 Neogoniolithon has an extremely long branch (as usual!). It may worth to try to remove it or to break the branch by adding some other species...

Line Le Gall

6. PLOS authors have the option to publish the peer review history of their article (what does this mean?). If published, this will include your full peer review and any attached files.

Reviewer #1: No

Reviewer #2: Yes: Line Le Gall

---

## [Author Response · Author response to Decision Letter 0]

15 Oct 2019

Response to reviewers and editors comments have been uploaded as a separate word document titled "Response to Reviewers"

---

## [Decision Letter · Decision Letter 1]

4 Nov 2019

PONE-D-19-23390R1

High diversity of coralline algae in New Zealand revealed: Knowledge gaps and implications for future research

PLOS ONE

Dear Dr. Twist,

Thank you for submitting your manuscript to PLOS ONE. After careful consideration, we feel that it has merit but does not fully meet PLOS ONE’s publication criteria as it currently stands. Therefore, we invite you to submit a revised version of the manuscript that addresses the points raised during the review process.

We would appreciate receiving your revised manuscript by Dec 19 2019 11:59PM. To enhance the reproducibility of your results, we recommend that if applicable you deposit your laboratory protocols in protocols.io, where a protocol can be assigned its own identifier (DOI) such that it can be cited independently in the future. For instructions see: http://journals.plos.org/plosone/s/submission-guidelines#loc-laboratory-protocols

We look forward to receiving your revised manuscript.

Kind regards,

Tzen-Yuh Chiang

Academic Editor

PLOS ONE

Reviewers' comments:

Reviewer's Responses to Questions

**Comments to the Author**

1. If the authors have adequately addressed your comments raised in a previous round of review and you feel that this manuscript is now acceptable for publication, you may indicate that here to bypass the “Comments to the Author” section, enter your conflict of interest statement in the “Confidential to Editor” section, and submit your "Accept" recommendation.

Reviewer #1: All comments have been addressed

Reviewer #2: All comments have been addressed

2. Is the manuscript technically sound, and do the data support the conclusions?

Reviewer #1: Yes

Reviewer #2: Yes

3. Has the statistical analysis been performed appropriately and rigorously? 

Reviewer #1: (No Response)

Reviewer #2: N/A

4. Have the authors made all data underlying the findings in their manuscript fully available?

Reviewer #1: (No Response)

Reviewer #2: Yes

5. Is the manuscript presented in an intelligible fashion and written in standard English?

Reviewer #1: (No Response)

Reviewer #2: Yes

6. Review Comments to the Author

Reviewer #1: Please add authors and adequate reference to previous taxonomic work throughout the manuscript. In particular:

Line 55 Add authorship of the three orders the first time you mention them

Line 94-95 Add authorship of these species the first time you mention in the manuscript: Lithophyllum kaiseri, Hydrolithon boergesenii, Sporolithon indopacificum

Line 97 add authorship for Phymatolithon the first time you mention in the manuscript

Line 213 Add authorship for Corallinapetra novaezelandiae the first time you mention in the manuscript

Line 271 Add authorship for Sporolithon, Heydrichia

Line 290 add authorship for Mastophora pacifica

Line 294 add authorship for Jania sagittata, Jania sphaeroramosa and to the other species mentioned on this same paragraph and in

Lines 509-510. The distribution of Sporolithon has been probably ruled by many other environmental controls in addition to temperature. It seems a deep genus in the tropics, but shallower in New Zealand as in the Red Sea, toward the limits of its distribution (Basso et al. 2009, Palaios), but it’s again deep in the Mediterranean. The maximum Late Cretaceous temperature is believed to be comparable to that of the PETM, but Sporolithon did not become abundant again during PETM. The picture is much more complicate than just temperature, and there is coralline evolution in between and increasing competition. I suggest to remove the comment about Sporolithon and the fossil record and the related reference.

Reviewer #2: I was very pleased to read the revised version of this manuscript. The authors addressed all my comments and I consider this new version ready to be published.

7. PLOS authors have the option to publish the peer review history of their article (what does this mean?). If published, this will include your full peer review and any attached files.

Reviewer #1: No

Reviewer #2: Yes: Line Le Gall

---

## [Author Response · Author response to Decision Letter 1]

4 Nov 2019

Dear Tzen-Yuh Chiang,

We are submitting a revised version of manuscript PONE-D-19-23390, in which we have made changes in response to the comments we received.

All changes that were suggested by reviewer 1 have been made. These include adding authorities to orders and species of corallines mentioned in the manuscript and removing the sentence relating to Sporolithon in the fossil record (lines 509-510). These changes are shown through the track change feature in MS word. A ‘clean’ version of this manuscript has also been submitted with all track changes accepted.

We would like to thank the reviewers for their constructive feedback and their time in reviewing the manuscript.

Sincerely,

Brenton and Co-authors

---

## [Decision Letter · Decision Letter 2]

11 Nov 2019

High diversity of coralline algae in New Zealand revealed: Knowledge gaps and implications for future research

PONE-D-19-23390R2

Dear Dr. Twist,

We are pleased to inform you that your manuscript has been judged scientifically suitable for publication and will be formally accepted for publication once it complies with all outstanding technical requirements.

With kind regards,

Tzen-Yuh Chiang

Academic Editor

PLOS ONE

Additional Editor Comments (optional):

Reviewers' comments:

Reviewer's Responses to Questions

**Comments to the Author**

1. If the authors have adequately addressed your comments raised in a previous round of review and you feel that this manuscript is now acceptable for publication, you may indicate that here to bypass the “Comments to the Author” section, enter your conflict of interest statement in the “Confidential to Editor” section, and submit your "Accept" recommendation.

Reviewer #2: All comments have been addressed

2. Is the manuscript technically sound, and do the data support the conclusions?

Reviewer #2: Yes

3. Has the statistical analysis been performed appropriately and rigorously? 

Reviewer #2: N/A

4. Have the authors made all data underlying the findings in their manuscript fully available?

Reviewer #2: Yes

5. Is the manuscript presented in an intelligible fashion and written in standard English?

Reviewer #2: Yes

6. Review Comments to the Author

Reviewer #2: This revised version reads very well. It is well written. The figures are very well presented. The results are new to science and will greatly contribute to highlight the great diversity of corallines species.

7. PLOS authors have the option to publish the peer review history of their article (what does this mean?). If published, this will include your full peer review and any attached files.

Reviewer #2: Yes: Line Le Gall

---

## [Editor Report · Acceptance letter]

15 Nov 2019

PONE-D-19-23390R2 

High diversity of coralline algae in New Zealand revealed: Knowledge gaps and implications for future research 

Dear Dr. Twist:

I am pleased to inform you that your manuscript has been deemed suitable for publication in PLOS ONE. Congratulations! Your manuscript is now with our production department. 

With kind regards,

on behalf of

Dr. Tzen-Yuh Chiang 

Academic Editor

PLOS ONE